# Metabolic Syndrome and Its Associated Early-Life Factors among Chinese and Spanish Adolescents: A Pilot Study

**DOI:** 10.3390/nu11071568

**Published:** 2019-07-11

**Authors:** Jiao Wang, Javier S. Perona, Jacqueline Schmidt-RioValle, Yajun Chen, Jin Jing, Emilio González-Jiménez

**Affiliations:** 1Department of Maternal and Child Health Care, School of Public Health, Sun Yat-Sen University, Guangzhou 510080, China; 2Instituto de la Grasa (CSIC), Campus Universidad Pablo de Olavide, Edificio 46, 41013 Seville, Spain; 3Departamento de Enfermería, CTS-436 Adscrito al Centro de Investigación Mente, Cerebro y Comportamiento (CIMCYC), University of Granada, Av/Ilustración 60, 18016 Granada, Spain

**Keywords:** metabolic syndrome, adolescents, breastfeeding duration, birth weight

## Abstract

Metabolic syndrome (MetS) is a growing problem worldwide in adolescents. This study compared two sample populations of young people in Spain and China, and analyzed the association of birth weight and breastfeeding duration with MetS. A cross-sectional study was conducted in adolescents (10–15 years old); 1150 Chinese and 976 Spanish adolescents. The variables analyzed were anthropometric characteristics, biochemical markers, and demographic characteristics using the same methodology and data collection protocol. Also, birth weight and breastfeeding were retrospectively analyzed during the first year of life. The results showed statistically significant differences between the two groups in reference to body mass index (BMI), blood pressure, triglyceride, glucose, and high-density lipoprotein cholesterol (HDL-C) levels. The MetS prevalence was higher in Spanish adolescents (2.5%) than in the Chinese group (0.5%). Breastfeeding duration was inversely associated with hypertriglyceridemia, low HDL-C, and MetS, whereas higher birth weight was associated with hyperglycemia, low HDL-C, hypertriglyceridemia, and abdominal obesity. Spanish adolescents showed more altered MetS components, and consequently, a higher MetS prevalence than the Chinese adolescents. This made them more vulnerable to cardiometabolic risk. Our results highlight the need for interventions designed by health professionals, which would encourage pregnant women to breastfeed their children.

## 1. Introduction

Metabolic syndrome (MetS) involves a cluster of risk factors. Subjects with MetS typically have at least three of the following conditions; abdominal obesity, hyperglycemia, hypertension, hypertriglyceridemia, and low HDL-cholesterol (HDL-C) levels [1]. MetS has been found to be strongly associated with a higher risk of cardiovascular disease in adults [2,3]. Unfortunately, there is a lack of data regarding MetS and its associated factors in childhood and adolescence. This is particularly the case in emerging countries such as China [4].

The prevalence of MetS in adolescents is currently increasing throughout the world [5]. In a Spanish population of 976 adolescents, 10–15 years of age, González-Jiménez et al. [6] observed a MetS prevalence of 3.9% in girls and 5.4% in boys. In contrast, Xu et al. [7] obtained a prevalence of 0.8% in Chinese adolescents over 10 years old. Similarly, in a study of 1770 Chinese adolescents by Wang et al. [8], the overall prevalence was 1.1%, though boys were more affected by MetS than girls. In these three studies, MetS was defined following to the criteria proposed by the International Diabetes Federation (IDF) for children and adolescents. According to these studies carried out in China and Spain it might be expected that MetS would be less frequent in Chinese adolescents. However, to date there has been no comparative study of MetS in sample populations in Spain and China. 

It is crucial to identify the risk factors leading to the development of MetS in young people so that this disorder can be rapidly detected and prevented [9]. Accordingly, it has been suggested that a higher birth weight may be associated with an early development of insulin resistance and MetS [10]. An excessively short period (or absence) of breastfeeding during the first year of childhood seems to be another risk factor for MetS [11]. In González-Jiménez et al. [6] and Wang et al. [8], a high birth weight was significantly associated with MetS, whereas breastfeeding for longer than six months was inversely associated with the syndrome. Nevertheless, there is no consensus of opinion on this issue since the absence of conclusive results in Western and Chinese populations [12] has generated a certain controversy regarding the influence of these variables [13].

The initial hypothesis of this research was that the difference in MetS prevalence in China and Spain was probably related to differences in birth weight and breastfeeding duration. To confirm this hypothesis, our objectives were to compare the prevalence of MetS in Chinese and Spanish adolescents, and then analyze the associations of MetS with birth weight and breastfeeding duration (first year of life).

## 2. Materials and Methods

### 2.1. Study Design and Participants

Accordingly, we decided to conduct a comparative analysis of two cross-sectional studies. The first study was of a sample of Chinese adolescents, and the second study was of a sample of Spanish adolescents. In China, the research was carried out in the city of Guangzhou (southern China). Four districts were randomly selected, which comprised a total of ten schools. A letter of invitation was sent to the school principals along with an information sheet and an explanation of the research methodology and objectives. All parents, tutors, and legal guardians of the minors gave their written informed consent. The study had been previously approved by the Ethics Committee of the School of Public Health in Sun Yat-sen University. 

In each of the ten schools, two classes per course were randomly selected and invited to participate in the study. A total of 1150 adolescents (554 boys and 596 girls), 10–15 years of age, were finally recruited for the study. All participants had a medium socioeconomic status and the response rate was 87%. 

In Spain, the study was conducted in the province of Granada (southeastern Spain), a total of eighteen schools throughout the province were randomly selected. The study had been previously approved by the Board of Education of the Andalusian Regional Government (Granada Delegation), and authorized by the school directors. It had also been approved by the Ethics Committee of the University of Granada. All parents, tutors, and guardians had explicitly authorized the participation of their children.

In each of the schools, two classes per course were randomly selected and invited to participate in the study. Finally, a total of 976 adolescents, (457 boys and 519 girls), 10–15 years of age, were recruited. All of the subjects were Spanish and had a medium socioeconomic status. 

Both in the Chinese and Spanish groups, adolescents with serious physical handicaps or psychological conditions (e.g., congenital disorder and cognitive dysfunction) were excluded from the study. The flow diagram (Figure 1) describes the selection process followed in both countries.

All procedures used in this study were in accordance with the ethical standards of the institutional and/or national research committee and with the 1964 Helsinki declaration and its later amendments or comparable ethical standards. 

### 2.2. Data Collection and Measurements

The instrument used for data collection was a standard questionnaire elaborated ad hoc. The questionnaire was filled out during a face-to-face interview with one of the parents or guardians. The average questionnaire completion time was 20–25 min.

Questionnaire items focused on the collection of demographic data and information regarding birth weight and breastfeeding during the subjects’ first year of life. All of this information was collected retrospectively. Furthermore, in order to guarantee the accuracy of the answers, parents were asked to bring the Child Health Record of their son/daughter. This document had been previously filled out by health professionals and contained accurate birth-weight and breastfeeding information. The completed questionnaires were reviewed by trained staff and uploaded into the database.

All participants underwent a complete anthropometric evaluation in accordance with the guidelines of the International Society for the Advancement of Kinanthropometry [14]. The variables assessed were weight, height, waist circumference (WC), hip circumference (HC), and BMI (weight (kg)/height (m)^2^). In the Chinese sample, fasting body weight was measured to the nearest 0.1 kg on a double ruler scale (RGT-140, Wujin Hengqi Co. Ltd, Changzhou, China). During this assessment, participants were wearing light clothing and no shoes. In the Spanish sample, the subjects (also in light clothing and no shoes) were weighed on a self-calibrating Seca 861 Class (III) Digital Floor Scale (Hamburg, Germany) with a precision of up to 0.1 kg.

In the Chinese sample, height was measured to an accuracy of 1mm with a freestanding stadiometer mounted on a rigid tripod (GMCS-I; Xindong Huateng Sports Equipment Co. Ltd, Beijing, China) by trained interviewers following a standardized protocol. Participants were asked to stand erect with their back, buttocks, and heels in continuous contact with the vertical height rod of the stadiometer and their head orientated in the Frankfurt plane. The horizontal headpiece was then placed on top of their head to measure their height. The height of the Spanish children was measured with a Seca 214* portable stadiometer (seca gmbh & co., Hamburg, Germany), following the same procedure as in the Chinese study. 

WC was measured with a Seca automatic roll-up measuring tape (precision of 1 mm) using the horizontal plane midway between the lowest rib and the upper border of the iliac crest at the end of a normal inspiration/expiration. HC was measured at the maximum extension of the buttocks as viewed from the right side. The average of two consecutive measurements was the value used in the analyses. In both samples, waist-to-height ratio was calculated as WC (cm) divided by height (cm), whereas waist-to-hip ratio was calculated by dividing WC by HC. The corresponding intraobserver technical error (reliability) of the measurements was 0.95%.

In the Chinese sample, blood pressure levels were calculated with a previously calibrated aneroid sphygmomanometer and a Littmann^®^ stethoscope (3M Health Care, Saint Paul, MI, USA) after each participant had rested for at least 15 min in a sitting position, according to the BP measurement guidelines of the Subcommittee of Professional and Public Education of the American Heart Association Council on High Blood Pressure Research [15]. Diastolic pressure was defined as the point of disappearance of the Korotkoff sounds (fifth phase). Blood pressure was taken twice on the right arm with an appropriately sized cuff. The average of two readings obtained at a minimum interval of 5 min was recorded. In the Spanish sample, blood pressure levels were calculated with the same equipment and following the same procedure as the research team in China.

### 2.3. Serum Biochemical Examination

The biochemical variables analyzed were fasting glucose (FG), HDL-C, and triglycerides (TG). At 8 a.m., after a 12 h overnight fast, 10 mL of blood was extracted by venipuncture from the antecubital fossa of the right arm with a disposable vacuum blood collection tube. In the 4 hours after extraction, all samples were centrifuged at 3500 rpm for 15 min (Z400 K, Hermle, Wehingen, Germany). Red blood cells were separated and serum was finally frozen at −80 °C for its subsequent analysis. FG was measured with a colorimetric enzymatic method (GOD-PAP Method, Human Diagnostics, Germany). HDL-C and TG were also calculated by means of a colorimetric enzymatic method with an Olympus analyzer (GmbH company, Hamburg, Germany). The techniques and equipment employed in the biochemical analysis were the same in both samples. The precision performance of these assays was within the manufacturer’s specifications. In both populations, blood samples were taken at the educational center in a classroom especially designated for this purpose and on different days in order to guarantee the fasting of the participants. 

### 2.4. Diagnostic Criteria of Metabolic Syndrome according to the International Diabetes Federation, IDF

The definition of MetS was based on the criteria of the IDF, adapted for children and adolescents. These criteria were abdominal obesity (defined by WC adult ethnicity-specific cutoffs: ≥94 cm in men and ≥80 cm in women for the Spanish and ≥90 cm in men and ≥80 cm in women for the Chinese) and the presence of two or more clinical features, including TG ≥1.7 mmol/L, HDL-C <1.03 mmol/L, systolic blood pressure (SBP) ≥130 mmHg and/or diastolic blood pressure (DBP) ≥85 mmHg, and serum FG ≥5.6 mmol/L [16].

### 2.5. Statistical Analysis

Descriptive statistics were calculated for all of the variables, including continuous variables (reported as mean and standard deviation) and categorical variables (reported as number and percentage). A *p*-value of 0.05 and a power of 80% were used to determine sample sizes. The differences between sexes, ages, and countries for the variables studied were evaluated using Student’s *t*-test, ANOVA, nonparametric test, or the χ^2^ test, as appropriate. Pearson/Spearman’s correlation coefficients between MetS components and their associated factors were also calculated. In the total sample and separated sample, multivariable logistic regression analyses (Enter method) were employed to identify the relationship between MetS and its features in the form of outcomes (abdominal obesity, low HDL-C, hyperglycemia, hypertriglyceridemia, hypertension), and associated factors. The multivariable logistic analysis in the country of origin (China), breastfeeding duration (months), and birth weight (100 g) was sex-adjusted and age-adjusted to control the influence of puberty. The prevalence odds ratios and the corresponding 95% confidence intervals were calculated as well. All statistical analyses were performed with the statistical software package SPSS version 21.0 (IBM, Armonk, NY, USA). In this study, *p*-values of less than 0.05 were regarded as statistically significant.

## 3. Results

Table 1. shows the data collected from the participants, including demographic information, anthropometry, MetS features, and early-life factors. It was observed that Spanish children had higher mean values for height, weight, BMI, WC, TG, and SBP than Chinese children. Generally speaking, the Spanish sample had also been breastfed for a longer period of time in their first year of life (9 months on average).

Regarding the prevalence of MetS and its components, Table 2 shows statistically significant differences between the two cohorts. As can be observed, the Spanish adolescents had a higher prevalence of abdominal obesity, hyperglycemia, and hypertension. In contrast, Chinese adolescents were more prone to hypertriglyceridemia though TG mean values were higher in the Spanish cohort. Consequently, the prevalence of MetS was 2.5% in the Spanish adolescents in comparison to 0.5% in the Chinese group.

Generally, the correlations of breastfeeding duration or birth weight with MetS and its components (Table 3) differed in Chinese and Spanish individuals. In Chinese adolescents correlations were weak. In the case of the Spaniards, the duration of breastfeeding correlated positively with HDL-C (0.81) and negatively with FG (−0.89) and TG (−0.64). In contrast, birth weight correlated negatively with HDL-C (−0.58) and positively with FG (0.65) and TG (0.48). In the Chinese cohort, birth weight was found to be positively associated with WC (0.15), SBP (0.08), and DBP (0.07).

Table 4 shows the adjusted associations of Mets features with early life factors including breastfeeding duration and birth weight in Chinese and Spanish adolescents. In the Spanish adolescents, breastfeeding duration had stronger associations with low HDL-C (OR 0.18), hyperglycemia (OR 0.17), hypertriglyceridemia (OR 0.52), and MetS (0.62) than in the Chinese adolescents. However, birth weight had closer associations with abdominal obesity (OR 1.09) in Chinese subjects and closer associations with hyperglycemia (OR 6.65) in Spanish subjects.

Furthermore, the combined multivariable analysis of MetS and its components showed that girls were more likely to develop abdominal obesity than boys (Table 5). In addition, there was a strong association of MetS and its components with the participants’ country of origin. In general, Spanish participants were at greater risk of MetS or of the alteration of certain MetS components. In this regard, the Spaniards had a higher risk of hypertension (OR 45.05), hyperglycemia (OR 26.85), and MetS (OR 13.6) than the Chinese adolescents. For the early-life factors, breastfeeding duration was negatively associated with hypertriglyceridemia (OR 0.87), low HDL-C (OR 0.81), hyperglycemia (OR 0.60), and MetS (OR 0.74), whereas higher birth weight was positively associated with MetS components such as hyperglycemia (OR 1.96) and abdominal obesity (OR 1.15) in the total sample. We also performed analyses to assess the interaction of breastfeeding duration and the country of origin and birth weight with the country of origin, using China as reference. When the interaction of breastfeeding duration and the country of origin was assessed, we observed that the effect was significant for low HDL-C, hyperglycemia and hypertriglyceridemia. In contrast, the interaction of birth weight with the country of origin was significant only for hyperglycemia. The results of the sensitivity analysis showed that breastfeeding duration (longer than 6 months) was a protective factor for MetS (Appendix A).

## 4. Discussion

The results obtained showed that there were significant differences in the Chinese and Spanish samples in regard to most of the clinical features analyzed, whether anthropometric or metabolic. More specifically, Spanish adolescents were found to have considerably higher levels of TG and SBP in comparison to Chinese adolescents. In contrast, the children in the Chinese sample had lower BMI, WHR, and WC values. Generally speaking, these results suggest that the Chinese subjects were healthier than their Spanish counterparts, who had higher values for most of the clinical features [17,18]. This may be partially explained by the ethnic difference and different living environments. This evidently placed their health at risk and made them more vulnerable to cardiometabolic disorders [19]. These results indicate the need for an in-depth study of the lifestyle, and environmental factors affecting the population in China and Spain. Any or all of these could be factors that would explain the differences found between adolescents in the two countries.

Our results also showed differences between the Chinese and Spanish samples regarding MetS components. As in previous studies of the Spanish population [20] and based on the definition of MetS used, Spanish children and adolescents had higher levels of abdominal obesity, hyperglycemia, as well as a higher rate of hypertension when compared to the Chinese group. Nevertheless, the prevalence of MetS was still lower than that reported by Holst-Schumacher et al. [21] for adolescents in Costa Rica (5.6%) or that described by Alvarez et al. [22] for adolescents in Brazil (6%). The only variable that was higher in Chinese cohort was the value for hypertriglyceridemia, which was in agreement with Liang et al. [23] who studied another population of 976 Chinese adolescents.

Breastfeeding duration and birth weight correlated closely with the components of MetS. In this regard, Spanish schoolchildren had been breastfed for a longer time. This correlated positively with HDL-C levels and negatively with other variables such as FG, TG, and MetS. These findings suggest the potentially positive effect of breastfeeding as a way to prevent the development of metabolic disorders in young subjects. These results contrasted with Yakubov et al. [24], who found no association between breastfeeding duration and the development of MetS and/or impairments of its components. However, this lack of association could be explained by factors such as the small size of the sample, a possible selection bias of the subjects or the range of breastfeeding duration. 

In contrast, other studies did find a correlation between breastfeeding duration and a reduced incidence of MetS later in life [25,26]. Yet, it could be argued that since in the present study Spanish adolescents were breastfed for a longer time and had a higher prevalence of MetS compared to the Chinese group, breastfeeding would increase, rather than decrease, the risk of developing MetS. However, this assumption can not be made directly, due to the multifactorial nature of the metabolic syndrome. The inverse association between breastfeeding duration and MetS in childhood and adolescence has been observed elsewhere [12,27]. Indeed, in a previous study, we found a higher prevalence of MetS in young subjects that had not been breastfed as babies [6]. Moreover, Ekelund et al. [28] stated that the most important benefits of maternal breastfeeding in terms of MetS prevention were for those subjects who had been breastfed for more than 6 months.

In the present study, breastfeeding duration was the main factor affecting the differences observed in the risk of MetS features between Chinese and Spanish adolescents. In fact, the interaction between breastfeeding duration and country for MetS and its components resulted strongly significant. Chinese mother breastfed their infants for less than half of the time than did Spanish mothers but no differences were on served for the range (0–13 months). A survey carried out in the city of Guangzhou (the same city of origin of our study), revealed that Chinese mothers tend to stop breastfeeding early before the six months. The reasons given for breastfeeding cessation were, among others, insufficient milk supply, medical reasons, lactational factors, and return to work [29]. Regardless of their antenatal intention, women with higher BMI have a higher risk of early cessation of exclusive breastfeeding [30], which has been associated to anatomical and physiological issues, medical conditions, and sociocultural and psychological factors [31]. Unfortunately, we do not have data on the prevalence of overweight mothers at the time of pregnancy. In our study, birth weight correlated positively with FG, TG, and MetS in Spanish subjects, as well as with WC in Chinese subjects. These findings do not coincide with those in Dos Santos et al. [32], who studied a population of 172 adolescents in Brazil and found that birth weight was not a risk factor in the development of MetS during adolescence. This result was probably conditioned by the small size of the Brazilian cohort, a possible bias in the selection of participants, or methodological differences in the evaluation of adiposity. Nonetheless, Yuan et al. [33] did find an association between birth weight and disorders such as adolescent obesity and MetS. Their population sample was much larger and was composed of 16,580 Chinese children and adolescents, 7–17 years of age, which is consistent with the Chinese sample in our study. Strikingly, the results obtained by Yuan et al. [33] were more in consonance with the Spanish results in our study than with the Chinese ones.

The results of the multivariable analysis of early MetS predictors with MetS and MetS components showed that girls had a higher risk of abdominal obesity, which agrees with previous research [34,35]. The high prevalence of central obesity in Spanish and Chinese girls is a matter of concern, since according to Lee et al. [36], it increases the risk of morbidity and mortality at early ages. Additionally, MetS and its components showed a strong association with the participants’ country of origin. In other words, Spanish adolescents had a greater risk of MetS or the alteration of any of its components (e.g., hypertension or hyperglycemia) than the Chinese adolescents. 

These results also contrast with those of Haldar et al. [37], who reported that Asian adults who emigrated and lived in European countries were more apt to become obese than Caucasians, regardless of the degree of adiposity. However, this may very well be due to changes in their dietary habits. To explain the discrepancies between the Chinese and Spanish populations, it could be argued that Spaniards are not typical Caucasians in regard to metabolic disorders. In comparison to young people in other European countries, children and adolescents in Spain are more prone to be overweight, which means greater metabolic risk. Accordingly, the prevalence of MetS in Spanish adolescents is higher than the MetS prevalence in other European countries, such as Finland (2.1%), Greece (0.7%) [38], Denmark, Estonia, and Portugal (1.4%) [28]. Other studies also highlight differences between ethnic groups in the distribution of body fat, which signifies that they are at greater risk of developing cardiovascular and metabolic pathologies [39,40]. 

Interestingly, our study found that hypertriglyceridemia was strongly affected by the duration of breastfeeding (OR 0.87), regardless of the participants’ gender or country of origin. The association between breastfeeding in infancy and triglyceride concentrations later in life is still a matter of debate. The absence of an association was reported by Victoria et al. [41] in a Brazilian population of 18-year-old boys and by Lawlor et al. [42] in Estonian and Danish children and adolescents. In Lawlor et al. [42], BMI and TG values were similar to those obtained for the Chinese sample in our study, but lower than those for the Spanish group. More recently, Ramirez-Silva et al. [43] observed that children who had not been breastfed had higher TG levels at the age of 4, compared with those who had been partially or exclusively breastfed. There is a lack of consensus regarding the influence of birth weight on hyperglycemia. 

This research has a number of strengths as well as certain limitations. Without a doubt, one of its strengths is its pioneering nature. Our study focused on two populations of adolescents with very different cultural backgrounds and analyzed not only the prevalence of MetS, but also the association between MetS and early predictors such as breastfeeding duration (first year of life) and birth weight. Yet another strength is the size of the sample, which enhances the validity of the results and assures comparability in future studies. Nonetheless, an evident limitation of the study is its cross-sectional nature and the lack of information about the eating habits and physical activity of both populations. Finally, no data regarding pubertal status were collected or taken into account but two subjects were excluded as outliers with extremely high triglyceride levels. This may have been a factor in the differences between the two countries. The results should thus be interpreted with caution. 

## 5. Conclusions

In conclusion, our initial hypothesis was confirmed since Spanish adolescents showed a higher number of altered MetS components, and consequently a higher prevalence of MetS than Chinese adolescents. In addition, in the Spanish sample, breastfeeding duration and birth weight strongly correlated with MetS components in comparison to the Chinese group, where this was not the case. The Spanish adolescents also had a higher risk for hyperglycemia, hypertension, hypertriglyceridemia, and abdominal obesity. 

The results of this study should have an impact on clinical practice. It is advisable for healthcare professionals to have an in-depth understanding of all of the factors associated with MetS. This knowledge and awareness are crucial to the prevention of this disease in adolescents. Likewise, as reflected in our results, it is important for healthcare professionals to encourage pregnant women to breastfeed their babies. This would improve the metabolic status of the mothers and at the same time make their children less vulnerable to obesity, diabetes mellitus type 2, and MetS. This is especially true for Spain, as we found important differences in the prevalence of MetS and its components between Chinese and Spanish adolescents, which were importantly affected by differences in breastfeeding duration in both countries. For this reason, a clear priority for health professionals should be to encourage breastfeeding.

## Figures and Tables

**Figure 1 nutrients-11-01568-f001:**
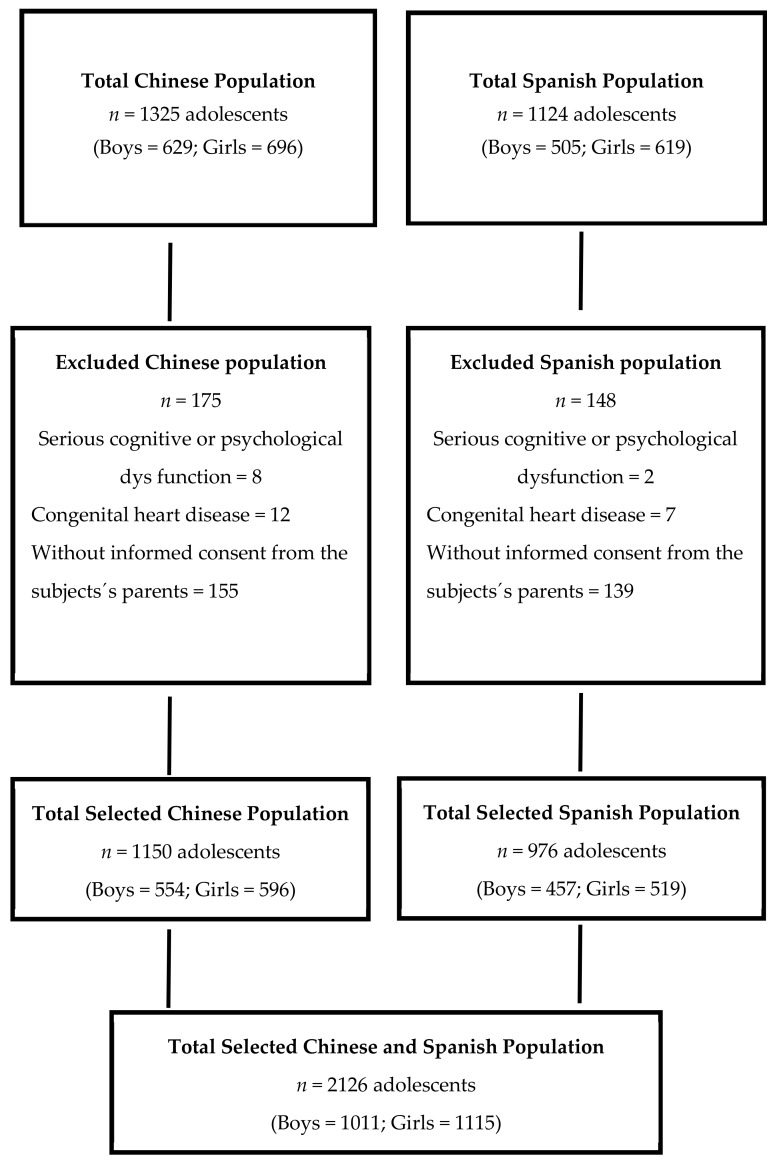
Flow diagram of the recruitment progress.

**Table 1 nutrients-11-01568-t001:** Demographic and clinical characteristics of Chinese and Spanish adolescents.

Variable	Chinese Adolescents	Spanish Adolescents	*p*-Value
Demographic information			
*n*	1150	976	
Boys (%)	554 (48.2)	457 (45.7)	0.28
Age (years)	12.9 ± 1.8	13.1 ± 1.2	0.09
Anthropometry			
Height (m)	1.55 ± 0.14	1.59 ± 0.10	<0.01 **
Weight (kg)	46.28 ± 14.01	54.77 ± 12.71	<0.01 **
BMI (kg/m^2^)	19.24 ± 7.25	21.22 ± 3.79	<0.01 **
HC (cm)	81.06 ± 16.91	83.82 ± 9.35	0.21
WHR	0.84 ± 0.09	0.87 ± 0.16	<0.01 **
WHtR	0.44 ± 0.07	0.45 ± 0.07	0.01 *
Metabolic syndrome features			
WC (cm)	66.7 ± 10.6	72.3 ± 10.8	<0.01 **
FG (mmol/L)	4.62 ± 0.49	4.77 ± 1.68	0.01 *
HDL-C (mmol/L)	1.4 ± 0.3	1.0 ± 0.1	<0.01 **
TG (mmol/L)	0.97 ± 0.51	1.43 ± 0.60	<0.01 **
SBP (mmHg)	99.9 ± 23.2	118.0 ± 15.5	<0.01 **
DBP (mmHg)	63.6 ± 15.7	64.1 ± 9.0	0.35
Information of early-life factors			
Breastfeeding duration (month)	4.2 ± 3.7	9.2 ± 2.7	<0.01 **
Breastfeeding duration range (month)	0~13	0~13	
Birth weight (kg)	3.23 ± 0.47	3.20 ± 0.50	0.09
Birth weight range (kg)	1.4~4.9	2.0~5.7	

Notes: BMI, body mass index; HC, hip circumference; WHR, waist-to-hip ratio; WHtR, waist-to-height ratio; WC, waist circumference; FG, fasting glucose; HDL-C, high-density lipoprotein cholesterol; TG, triglyceride; SBP, systolic blood pressure; DBP, diastolic blood pressure. Values are expressed as mean ± SD. The *p*-value was calculated by chi-square test for category variables and two-sample *t*-test for continuous variables. * *p* < 0.05, ** *p* < 0.01.

**Table 2 nutrients-11-01568-t002:** Prevalence of metabolic syndrome and its components in Chinese and Spanish adolescents.

Characteristics	Chinese Adolescents (*n* = 1150)	Spanish Adolescents (*n* = 976)
Abdominal obesity	65 (5.5)	128 (13.1) **
Hypertension	11 (1.0)	256 (26.2) **
Hyperglycemia	23 (2.0)	65 (6.7) **
Low HDL-C	160 (13.9)	176 (18.0) *
Hypertriglyceridemia	84 (7.3)	32 (3.3) **
Pattern of risk factors clustering		
0 Component	886 (77.0)	568 (58.2) **
1 Component	193 (16.8)	234 (24.0) **
2 Components	61 (5.6)	120 (12.3) **
≥3 Components	7 (0.6)	54 (5.5) **
MetS	6 (0.5)	24 (2.5) **

Notes: HDL-C, high-density lipoprotein cholesterol; MetS, metabolic syndrome. Values are expressed as numbers of individuals (%). *p*-value was calculated by chi-square; * *p* < 0.05, ** *p* < 0.01.

**Table 3 nutrients-11-01568-t003:** The association of metabolic syndrome features and early-life factors in Chinese and Spanish adolescents.

	Chinese Adolescents (*n* = 1150)		Spanish Adolescents (*n* = 976)	
	Breastfeeding Duration	Birth Weight	Breastfeeding Duration	Birth Weight
WC	−0.04	0.15 **	0.05	0.10 **
FG	0.03	0.01	−0.89 **	0.65 **
HDL-C	−0.04	−0.01	0.81 **	−0.58 **
TG	0.09 **	−0.04	−0.64 **	0.48 **
SBP	0.03	0.08 *	−0.03	0.06
DBP	−0.02	0.07 *	−0.02	0.07 *

Notes: WC, waist circumference; FG, fasting glucose; HDL-C, high-density lipoprotein cholesterol; TG, triglyceride; SBP, systolic blood pressure; DBP, diastolic blood pressure; MetS, metabolic syndrome. Data are presented as Pearson/Spearman’s correlation coefficients. * *p* < 0.05 ** *p* < 0.01.

**Table 4 nutrients-11-01568-t004:** Risk of MetS features based on associated factors from multivariable logistic regression in two sample.

	Chinese Adolescents (*n* = 1150)		Spanish Adolescents (*n* = 976)	
	Breastfeeding Duration (month)	Birth Weight (100 g)	Breastfeeding Duration (month)	Birth Weight (100 g)
	OR (95% CI)	OR (95% CI)	OR (95% CI)	OR (95% CI)
Abdominal obesity	0.95 (0.88, 1.02)	1.09 (1.03, 1.16)	0.98 (0.90, 1.07)	1.04 (0.99, 1.10)
Low HDL-C	1.03 (0.98, 1.08)	1.17 (0.98, 1.06)	0.18 (0.13, 0.26)	1.06 (0.93, 1.21)
Hyperglycemia	0.96 (0.86, 1.08)	0.96 (0.88, 1.05)	0.17 (0.04, 0.72)	6.65 (1.83, 24.19)
Hypertriglyceridemia	1.03 (0.97, 1.10)	0.99 (0.94, 1.04)	0.52 (0.40, 0.70)	1.09 (0.93, 1.27)
Hypertension	1.10 (0.91, 1.32)	1.07 (0.92, 1.23)	0.96 (0.90, 1.03)	1.02 (0.99, 1.07)
MetS	1.07 (0.87, 1.32)	0.93 (0.80, 1.08)	0.62 (0.51, 0.76)	0.98 (0.86, 1.12)

Note: Age and gender were adjusted in all above models.

**Table 5 nutrients-11-01568-t005:** Risk of MetS features based on associated factors from multivariable logistic regression.

Variables (Reference)	Abdominal Obesity	Low HDL-C	Hyperglycemia	Hypertriglyceridemia	Hypertension	MetS
	OR (95% CI)	OR (95% CI)	OR (95% CI)	OR (95% CI)	OR (95% CI)	OR (95% CI)
Age	1.20 (1.09, 1.32)	1.07 (0.99, 1.15)	0.96 (0.80, 1.16)	0.92 (0.83, 1.02)	1.04 (0.93, 1.17)	1.01 (0.80, 1.28)
Gender (Boy)	2.40 (1.72, 3.35)	0.78 (0.58,1.07)	0.30 (0.12, 0.73)	0.79 (0.53, 1.20)	0.77 (0.58, 1.02)	1.76 (0.79, 3.89)
Country (China)	12.04 (0.58, 24.23)	3.84 (2.68, 5.49)	26.85 (11.76, 61.34)	0.70 (0.43, 1.15)	45.05 (21.71, 93.45)	13.6 (4.16, 44.54)
Breastfeeding duration (month)	0.91 (0.77, 1.08)	0.81 (0.78, 0.85)	0.60 (0.53, 0.68)	0.87 (0.82, 0.92)	1.10 (0.91, 1.32)	0.74 (0.65, 0.84)
Birth weight (100 g)	1.15 (1.01, 1.31)	0.98 (0.84, 1.10)	1.96 (1.88, 3.08)	0.99 (0.95, 1.04)	1.07 (0.92, 1.23)	0.94 (0.80, 1.09)
	*p* value	*p* value	*p* value	*p* value	*p* value	*p* value
Breastfeeding duration*Country	0.504	<0.001	<0.017	<0.001	0.187	<0.001
Birth weight*Country	0.211	0.578	<0.001	0.213	0.605	0.743

Notes: HDL-C, high-density lipoprotein cholesterol; MetS, metabolic syndrome. Data are presented as prevalence odds ratio (OR) with 95% confidence intervals (CI) using logistic regression model. Bold type indicates *p* < 0.05. All the shown risk factors in this model were included in the model at the same time.

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
