# Peer review of "Metabolic Syndrome and Its Associated Early-Life Factors among Chinese and Spanish Adolescents: A Pilot Study"

_nutrients, 2019, doi:10.3390/nu11071568_

Round 1
Reviewer 1 Report
A multi-ethnic study could provide valuable information. However, the results from two nations could not be generalized to adolescents worldwide. Is there cooperation of health programs? Are the sizes of statistical parent populations assumed from samples China:Spain = 1:1? This paper has other methodological problems.
Line 36
The word following "by" should be a subject of an active sentence.
Lines 38-41
An adversative conjunction "however" links an outcome of MetS to a cause of MetS. The authors should be stated the effect of adolescent MetS as first.
Lines 42-48
The authors stated that "the prevalence is increasing," but the prevalence at one point were described, instead of changes of the prevalence. Although the authors defined MetS as at least three conditions (Lines 36-38), they referred the prevalence of at least one factor {Ref 5].
Line 51
Does this sentence means that "they are less likely to become MetS because they are less vulnerable to obesity?" A cause of outcome must be placed after "vulnerable to."
Same to Line 315
Line 54
Is Reference 9, which stated the cross-sectional prevalence, correct?
Lines 69-
Ref. 9 demonstrated discrepant risk patterns observed between urban and rural areas. Are the subjects and study areas representatives of eath nation/ethnicity?
Repetition in Lines 81-82, and Lines 90-92. The same description for both areas should be omitted.
Line 92
Figure 1 is not found.
Line 156-158
"Gender-, and ethnicity-specific cut-offs" are preposition to a single definition for adults in Ref. 16. Although it is also stated that values above the 90th percentiles for gender and age are used in children (P301 in Ref. 16), percentiles rather than absolute values of WC have been used in the new criteria to compensate for varying degrees of development and ethnicity in the youth population. (P303 in Ref. 16)
These sentences indicates that when comparing different ethnicity population like this study, use each percentile cutoff (age-, and gender-specific) of each ethnicity before comparing, so a proportion value based on a percentile definition must be similar among different ethnicity. However, the prevalence of abdominal obesity was different between two groups. (Lines 187-188, and Table 2) This concern is important to answer the first object of this study, to "compare the prevalence of MetS in Chinese and Spanish adolescents." Indicate cutoffs.
The second object of this study was “to analyze the associations of MetS with birth weight and breastfeeding duration (first year of life).” (Lines 65-66)
It is unknown that two groups could be combined in one model. Two groups are different in some traits, and an effect of breastfeeding might appear in a part of range of obesity, instead of all range. Therefore, first, two groups should be analyzed separately in multivariable models. Considering the results of separate models, second, combined groups are analyzed using a mixed model (like meta-analysis), or interaction terms are included in a multivariate model.
Table 1. SDs of weight, BMI, HC, FG, SBP, and DBP are different between two groups, regardless of about 1000 subjects. How did the authors treat outliers, and check the methods, and did they compare the results from the subjects with and without markedly high/low values (or outliers). Discuss these differences of SDs.
Line 187-188.
higher values -> higher prevalence?
Line 188-189
"In contrast, Chinese adolescents were more prone to hypertriglyceridemia though TG mean values were higher in the Spanish cohort."
This pattern casts doubt that some Chinese adolescents took breakfast before blood sampling. Two hours after breakfast, blood glucose returns to a normal level, but triglyceride keeps a high level until 4 hours after breakfast. East Asian and Latin European have different breakfast skipping rates, and different weight to breakfast (PMC6165504, PMID: 27472357 PMCID: PMC4997447 DOI: 10.3390/ijerph13080761, and PMID: 28592097 DOI: 10.3760/cma.j.issn.0253-9624.2017.06.013). Furthermore, adolescents, due to their self-assertion, are likely not to comply with instruction. Blood was sampled "at 8 a.m., after a 12-h overnight fast," (Line 142) which is a routine statement, but it is difficult to withdraw blood from all students of two classes at one point time. How did the authors instruct and ascertain fasting status of the participants? Describe blood sampling methods in detail, such as time (to know time after school start), place (school?), season, and year. Did the authors exclude the participants who took breakfast before blood sampling, or how did they do? When the information about breakfast taking would be incorrect (possibility), the subjects with high TG (defining cutoffs in this study) should be excluded. If necessary, results of a sensitivity analysis should be indicated.
Line 195
“Correlations between breastfeeding duration and birth weight with MetS and its components 195 (Table 3) were weaker for Chinese adolescents.” However, effect sizes (correlation coefficients) in Chinese adolescents are larger than those in Spanish. WC, and SBP for birth weight.
Line 198, and Table 3.
What type of variables is "MetS" for correlations? binary, ordered, or numerical?
Table 3
The unit of birth weight is g, instead of kg in other places. Use same units throughout the manuscript.
Lines 195-196
Correlations between breastfeeding duration and birth weight with MetS and its components 195 (Table 3)
-> Correlations of breastfeeding duration or birth weight with MetS and its components?
Line 211-212
“Breastfeeding duration was associated with hypertriglyceridemia (OR 0.87), low HDL-C (OR 0.81), and MetS (OR 0.74)”
Positive or negative associations? A long duration is associated with a low or high prevalence?
Table 4
When odds ratios are calculated, unit, kg, is inappropriate to public health fields, because SDs are near 0.5. It is also said about breastfeeding duration.
Lines 254-256
The report of Ekelund et al. (Ref 29) indicated a non-linear association, instead of a linear association? Reconsider analytic models, or discuss it.
Lines 247-250
Who argues that breastfeeding increase the risk of MetS?
Lines 244-246, and 258-260
The authors of Refs. 24, and 30. did not find significant associations, but they did not always deny the associations, because of a small sample size, selection bias of the subjects, the extent of adiposity, or the range of breastfeeding duration. Discuss each reference in those viewpoints, and consider results of multivariate analysis of separated groups, Spain and China, and possibility of non-linear association.
Line 300
“the lack of information about the eating habits and physical activity of both populations.”
What should be cautioned? How about genetic traits (ethnicity)?
Line 306-307
“in the Spanish sample, breastfeeding duration and birth weight strongly correlated with MetS components in comparison to the Chinese group”
How did the authors compare two groups?
Line 316
“A clear priority for health professionals should be to encourage breastfeeding and foment healthy eating habits.”
Is this sentence applied only to Spanish population, or both populations? Is this conclusion able to apply to Chinese, although breastfeeding duration in Chinese population was shorter than those in Spanish? What result implies the recommendation “to foment healthy eating habits?” The authors wrote lack of information about eating habits in Line 300.
Author Response
Reviewer 1
A multi-ethnic study could provide valuable information. However, the results from two nations could not be generalized to adolescents worldwide. Is there cooperation of health programs?
We would like to highlight the fact that these are two independent studies carried out in Spain and China. They are the result of an exploratory international collaboration between researchers in these countries. In the future, it is our hope to further expand this coordinated research work so that it will eventually lead to joint health programs and international projects between Spain and China.
Are the sizes of statistical parent populations assumed from samples China: Spain = 1:1? This paper has other methodological problems.
The sample populations are of similar size (1150 Chinese adolescents and 976 Spanish adolescents). Their sex, age, birth weight and socioeconomic level were also similar. The samples studied are not a reflection of the totality of the adolescent population in Spain and China. To make this clearer, the title has been modified as follows:
“Metabolic syndrome and its associated early-life factors among Chinese and Spanish adolescents: A pilot study”
Line 38-99: The word following "by" should be a subject of an active sentence.
The sentence has been revised and is now in the active voice.
Lines 40-43: An adversative conjunction "however" links an outcome of MetS to a cause of MetS. The authors should be stated the effect of adolescent MetS as first.
The sentence has been rephrased as suggested by the reviewer.
Lines 44-51: The authors stated that "the prevalence is increasing," but the prevalence at one point were described, instead of changes of the prevalence. Although the authors defined MetS as at least three conditions (Lines 36-38), they referred the prevalence of at least one factor {Ref 5].
Thank you for your observation. The quotation from Miller et al. (ref.5) has been revised as follows (page 2, Line 46):
“The estimated metabolic syndrome prevalence was 10.1%.”.
Lines 52-53: Does this sentence means that "they are less likely to become MetS because they are less vulnerable to obesity?" A cause of outcome must be placed after "vulnerable to."
Since the sentence was superfluous, it has been eliminated:
“This seems to indicate that they are less vulnerable to MetS than adolescents in Western countries, such as Spain”
Same to Line 315
This line highlights the importance of maternal breastfeeding as a factor that protects against obesity, diabetes mellitus type 2, and MetS.
Line 56: Is Reference 9, which stated the cross-sectional prevalence, correct?
Yes. it is.
Lines 71: Ref. 9 demonstrated discrepant risk patterns observed between urban and rural areas. Are the subjects and study areas representatives of eath nation/ethnicity?
These are two independent transversal studies conducted in China and Spain. The samples are not representative of the total population in these countries even though the subjects are of similar sex, age, birthweight, and socioeconomic level.
Repetition in Lines 82-83, and Lines 92-93. The same description for both areas should be omitted.
Lines 82-83 refer to the study of Chinese adolescents, and lines 92-93 refer to the study of Spanish adolescents.
Line 98: Figure 1 is not found.
We have rectified the error. Figure 1 is now included in the text.
Line 156-158 (now lines 167-173): "Gender-, and ethnicity-specific cut-offs" are preposition to a single definition for adults in Ref. 16. Although it is also stated that values above the 90th percentiles for gender and age are used in children (P301 in Ref. 16), percentiles rather than absolute values of WC have been used in the new criteria to compensate for varying degrees of development and ethnicity in the youth population. (P303 in Ref. 16)
These sentences indicates that when comparing different ethnicity population like this study, use each percentile cutoff (age-, and gender-specific) of each ethnicity before comparing, so a proportion value based on a percentile definition must be similar among different ethnicity. However, the prevalence of abdominal obesity was different between two groups. (Lines 187-188, and Table 2) This concern is important to answer the first object of this study, to "compare the prevalence of MetS in Chinese and Spanish adolescents." Indicate cutoffs.
Thank you for raising this point. Actually, we used the ethnicity-specific adult waist circumference cutoffs to define central obesity in the adolescents for age 10 to<16 years(1). The ethnicity-specific waist circumference cutoffs are applicable to different ethnic groups. It allowed us to compare the prevalence of abdominal obesity and MetS in Chinese and Spanish. The methods are now described as follows:
“These criteria were abdominal obesity (defined by WC adult ethnicity specific cutoffs) and the presence of two or more clinical features, including TG ≥ 1.7 mmol/l, HDL-C < 1.03 mmol/l, systolic blood pressure (SBP) ≥ 130 mmHg and/or diastolic blood pressure (DBP) ≥ 85 mmHg, and serum FG ≥ 5.6 mmol/l [16]. “
The second object of this study was “to analyze the associations of MetS with birth weight and breastfeeding duration (first year of life).” (Lines 65-66) (now lines 67-68)
It is unknown that two groups could be combined in one model. Two groups are different in some traits, and an effect of breastfeeding might appear in a part of range of obesity, instead of all range. Therefore, first, two groups should be analyzed separately in multivariable models. Considering the results of separate models, second, combined groups are analyzed using a mixed model (like meta-analysis), or interaction terms are included in a multivariate model.
Following the reviewer’s suggestions, we first analyzed the two groups using the multivariable models separately. Breastfeeding duration and birth weight were found to have different effects on Chinese and Spanish adolescents. We then combined the two groups using the multivariate model with interaction terms for Breastfeeding duration*Country and Birth weight*Country. We added Table 4 and modified Table 5 (original Table 4). The methods and results were revised as follows:
Methods:
“In the total sample and separated sample...”
Results:
Table 1. SDs of weight, BMI, HC, FG, SBP, and DBP are different between two groups, regardless of about 1000 subjects. How did the authors treat outliers, and check the methods, and did they compare the results from the subjects with and without markedly high/low values (or outliers). Discuss these differences of SDs.
In accordance with the reviewer’s observations, some outliers were excluded previous to the analysis. This was done by using the Pauta criterion, and excluding those who reported weight, BMI, HC, FG, SBP, or DBP >±3SD from the mean. However, there was still a considerable difference in the SDs of the two groups, which could be partly explained by the ethnic difference and different living environments. This is now stated in the manuscript.
Discussion:
“Generally speaking, these results suggest that the Chinese subjects were healthier than their Spanish counterparts, who had higher values for most of the clinical features [17,18]. This may be partially explained by the ethnic difference and different living environments.”
Line 187-188: higher values -> higher prevalence? (now 204-205)
This sentence has been revised as suggested by the reviewer
“As can be observed, the Spanish adolescents had a higher prevalence of abdominal obesity...”
Line 188-189 (now 204-206):
"In contrast, Chinese adolescents were more prone to hypertriglyceridemia though TG mean values were higher in the Spanish cohort."
This pattern casts doubt that some Chinese adolescents took breakfast before blood sampling. Two hours after breakfast, blood glucose returns to a normal level, but triglyceride keeps a high level until 4 hours after breakfast. East Asian and Latin European have different breakfast skipping rates, and different weight to breakfast (PMC6165504, PMID: 27472357 PMCID: PMC4997447 DOI: 10.3390/ijerph13080761, and PMID: 28592097 DOI: 10.3760/cma.j.issn.0253-9624.2017.06.013). Furthermore, adolescents, due to their self-assertion, are likely not to comply with instruction. Blood was sampled "at 8 a.m., after a 12-h overnight fast," (Line 142) which is a routine statement, but it is difficult to withdraw blood from all students of two classes at one point time. How did the authors instruct and ascertain fasting status of the participants? Describe blood sampling methods in detail, such as time (to know time after school start), place (school?), season, and year. Did the authors exclude the participants who took breakfast before blood sampling, or how did they do? When the information about breakfast taking would be incorrect (possibility), the subjects with high TG (defining cutoffs in this study) should be excluded. If necessary, results of a sensitivity analysis should be indicated.
We understand the reviewer’s observations, but since these subjects were minors, parents or guardians informed the researchers in situ that their children had fasted before the blood sampling. Blood samples were taken at the educational center in a classroom especially designated for this purpose and on different days in order to guarantee fasting. In all cases, blood samples were taken before clases began so that after the extraction, adolescents could have breakfast in the school cafetería.
Line 195 (now 2012) : “Correlations between breastfeeding duration and birth weight with MetS and its components 195 (Table 3) were weaker for Chinese adolescents.” However, effect sizes (correlation coefficients) in Chinese adolescents are larger than those in Spanish. WC, and SBP for birth weight.
The positive association of WC with birth weight was more pronounced in Chinese adolescents than in Spansh adolescents (Table 3). This was also consistent with the separate multivariable logistic regression results (Table 4). We have modified this sentence as follows
“Generally, the correlations between breastfeeding duration and birth weight with MetS and its components (Table 3) were weaker for Chinese adolescents. However, the Chinese WC and SBP showed stronger associations with birth weight.”
Line 198 (now line 222), and Table 3:
What type of variables is "MetS" for correlations? binary, ordered, or numerical?
The variables are binary.
Table 3: The unit of birth weight is g, instead of kg in other places. Use same units throughout the manuscript.
The Pearson correlation coefficient was used to assess the linear correlation between the metabolic syndrome features and early-life factors in Chinese and Spanish adolescents in
Table 3. According to the formula: , the value of the coefficient is not relevant to the unit of variables X and Y. We thus deleted all the units in Table 3. For the unit in Tables 4 and 5, we changed the birth weight unit from kg to 100g because grams are too small to evaluate the regression association and kilograms are too large. We only maintained the kg in Table 1 to make the description clearer.
Lines 195-196 (now line 212):
Correlations between breastfeeding duration and birth weight with MetS and its components 195 (Table 3)
Correlations of breastfeeding duration or birth weight with MetS and its components?
This sentence has now been modified as follows:
“Generally, the correlations between breastfeeding duration and birth weight with MetS and its components (Table 3) were weaker for Chinese adolescents. However, the Chinese WC and SBP showed stronger associations with birth weight.”
Line 211-212 (now lines 244-246)
“Breastfeeding duration was associated with hypertriglyceridemia (OR 0.87), low HDL-C (OR 0.81), and MetS (OR 0.74)”
Positive or negative associations? A long duration is associated with a low or high prevalence?
The association is negative. The OR represents the odds that an outcome will occur given a particular exposure as compared to the odds of the outcome occurring in the absence of that exposure. OR=1 means that breastfeeding duration does not affect Mets; OR>1 means long duration is associated with a higher prevalence; and OR<1 means long duration is associated with a lower prevalence.
Table 4: When odds ratios are calculated, unit, kg, is inappropriate to public health fields, because SDs are near 0.5. It is also said about breastfeeding duration.
As suggested by the reviewer, we have changed the birth weight unit from kg to 100g in Tables 4 and 5. However, the breastfeeding duration (in months) may be applicable here because it has been widely used in other papers(2) and was more easily accepted by participants in our cohort.
Lines 254-256 (now lines 290-292):
The report of Ekelund et al. (Ref 29) indicated a non-linear association, instead of a linear association? Reconsider analytic models, or discuss it.
We performed a sensitivity analysis in which breastfeeding duration was regarded as a categorical variable (duration longer than 6 months and shorter than 6 months), but the results were not consistent. Longer breastfeeding duration was associated with a lower MetS prevalence. The sensitivity analysis results have been included in the manuscript.
Results (Lines 249-251):
“The results of a sensitivity analysis showed that breastfeeding duration (longer than 6 months) was a protective factor for MetS (Supplementary Table 1).”
Lines 247-250 (now 284-288): Who argues that breastfeeding increase the risk of MetS?
This is the tentative conjecture of the authors, based on the results obtained. Nevertheless, the following sentence clarifies that this cannot be assumed: “However, this assumption can not be made directly, due to the multifactorial nature of the metabolic syndrome”.
Lines 244-246 (now lines 281-282), and 258-260 (now lines 296-298): The authors of Refs. 24, and 30. did not find significant associations, but they did not always deny the associations, because of a small sample size, selection bias of the subjects, the extent of adiposity, or the range of breastfeeding duration. Discuss each reference in those viewpoints, and consider results of multivariate analysis of separated groups, Spain and China, and possibility of non-linear association.
In response to the reviewer’s suggestion, we have expanded and improved the discussion of our results in relation to references 24 and 30. We have added a separate analysis for each group. We also performed the non-linear logistic regression in the form of a sensitivity analysis.
Line 300 (now 337-340): “the lack of information about the eating habits and physical activity of both populations.” What should be cautioned? How about genetic traits (ethnicity)?
The lack of information about eating habits and physical activity has been included as one of the limitations of this study (page 11, lines 337-340). For this reason, our results should be interpreted with caution.
Although the consideration of ethnic characteristics was not one of the objectives of this study, it will be considered in future studies.
Line 306-307 (now lines 344-345):
“in the Spanish sample, breastfeeding duration and birth weight strongly correlated with MetS components in comparison to the Chinese group”. How did the authors compare two groups?
Yes, a clearer explanation of the comparison of the two groups is necesssary. As reflected in Tables 3 and 4, breastfeeding duration was more strongly associated with Low HDL-C, Hyperglycemia, Hypertriglyceridemia and MetS in the Spanish cohort than in the Chinese cohort. However, birth weight was more closely related to abdominal obesity in the Chinese sample and with hyperglycemia in the Spanish sample. We have addaed the new results as follows:
Results:
“Table 4 shows the adjusted associations of Mets features with early life factors including breastfeeding duration and birth weight in Chinese and Spanish adolescents. In the Spanish adolescents, breastfeeding duration had stronger associations with Low HDL-C (OR 0.18), Hyperglycemia (OR 0.17), Hypertriglyceridemia (OR 0.52) and MetS (0.62) than in the Chinese adolescents. However, birth weight had closer associations with abdominal obesity (OR 1.09) in Chinese subjects and closer associations with Hyperglycemia (OR 6.65) in Spanish subjects.”
Line 316 (now lines 354-355):
“A clear priority for health professionals should be to encourage breastfeeding and foment healthy eating habits.”
Is this sentence applied only to Spanish population, or both populations?
Yes, it is applicable to both populations.
Is this conclusion able to apply to Chinese, although breastfeeding duration in Chinese population was shorter than those in Spanish? What result implies the recommendation “to foment healthy eating habits?” The authors wrote lack of information about eating habits in Line 300.
Yes, it is applicable to both populations.
In line with the reviewer’s suggestion, we have eliminated the sentence “…and foment healthy eating habits”. (Page 12, line 356).
Reviewer 2 Report
The authors undertook a cross-sectional analysis of the prevalence of metabolic syndrome and its individual components in Spanish and Chinese adolescents, comparing both countries and relating the clinical outputs with breastfeeding duration and body weight at birth. The study is of great interest to the scientific community and it was very well addressed and presented in the manuscript. I found it very interesting to read. Despite some limitations, it may be a step towards further research into metabolic abnormalities in this under-researched population (adolescents). I have, however, some comments and suggestions to make, as follows:
1) The greater limitations of the study were:
a) the lack of nutritional. A food frequency questionnaire would be highly relevant. We are talking about 2 completely different populations, with different dietary patterns. Knowing how diet influences the development of MetS, it is critical to address food intake in these types of studies.
b) Lack of physical activity information. Exercise is one of the most critical factors influencing insulin sensitivity and the entire MetS is based on the pre-assumption that there is some degree of insulin resistance, so it seems important to address this.
c) Spanish participants were taller, heavier, etc. The majority was probably at a different stage of development/puberty when compared to the Chinese ones.
2) It hasn't been described how many had LOW birth weight (<2,5kg) or="" high="" birth="" macrossomia="">4kg). This is absolutely relevant and the data should be adjusted for these confounding factors. If many of the Spanish children were born with low or high body weight, those could be contributing to greater prevalence of MetS in these teenagers. I would do a sensitivity analysis addressing these parameters, if the data is available.
3) How many were preterm babies? Do you have that info? I also find this to be relevant, although not as important as the high or low birth weight. Generally preterm babies are born with low birth weight anyway, and this has an effect on the development of metabolic abnormalities later in life.
4) In table 2, you presented numbers of participants who had 0, 1, 2 or 3 or more components of the MetS and then those who had MetS... I am confused because, based on the paper published in circulation in 2009 (Alberti et al.) harmonizing the metabolic syndrome, it has been agreed that ANYONE having 3 or more components IS classified as having MetS. How is it possible that you have different numbers for those having 3 or more components and those with a full diagnosis of MetS??? Is this different in adolescents somehow??
5) Finally, you discuss how there are lower incidences (OR) of hypertriglyceridemia in those being breastfed but, in one of the erlier tables, you clearly show a positive correlation in Chinese children between TGs and being breastfed (statistically significant). How do you interpret the contradictory results?
Minor comments:
In the method section (2.2), lines 111 and 115, you describe the Chinese and Spanish methodology for measuring body weight; in one you say to the nearest 0.1kg and the other you say to the nearest 100g. This is the same, so I suggest keeping consistency.
Author Response
Reviewer 2
The authors undertook a cross-sectional analysis of the prevalence of metabolic syndrome and its individual components in Spanish and Chinese adolescents, comparing both countries and relating the clinical outputs with breastfeeding duration and body weight at birth. The study is of great interest to the scientific community and it was very well addressed and presented in the manuscript. I found it very interesting to read. Despite some limitations, it may be a step towards further research into metabolic abnormalities in this under-researched population (adolescents). I have, however, some comments and suggestions to make, as follows:
We greatly appreciate your kind words. However, we would like to point out that this paper describes two independent studies carried out in Spain and China. We were thus only able to compare the variables available in these studies. This research is a preliminary collaboration between researchers in both countries.
1) The greater limitations of the study were:
a) the lack of nutritional. A food frequency questionnaire would be highly relevant. We are talking about 2 completely different populations, with different dietary patterns. Knowing how diet influences the development of MetS, it is critical to address food intake in these types of studies.
We agree that nutritional data would have greatly enhanced our research. However, since this is a pilot study and only an initial approach to the research question, dietary information was finally not collected. The lack of nutritional data has now been included as a limitation of this study. (page 11, lines 337-340).
b) Lack of physical activity information. Exercise is one of the most critical factors influencing insulin sensitivity and the entire MetS is based on the pre-assumption that there is some degree of insulin resistance, so it seems important to address this.
We also agree that information regarding the physical activity of the participants would have been very relevant. However, in this preliminary stage of our research, we still have not obtained this information. Nevertheless, the lack of these data has been included as a limitation of this study (page 11, lines 337-340).
c) Spanish participants were taller, heavier, etc. The majority was probably at a different stage of development/puberty when compared to the Chinese ones.
We understand your observation, but unfortunately, we were not authorized to collect that information in both populations. This has also been included as a limitation of this study. (page 11, lines 337-340).
2) It hasn't been described how many had LOW birth weight (<2,5kg) or="" high="" birth="" macrossomia="">4kg). This is absolutely relevant and the data should be adjusted for these confounding factors. If many of the Spanish children were born with low or high body weight, those could be contributing to greater prevalence of MetS in these teenagers. I would do a sensitivity analysis addressing these parameters, if the data is available.
The categorical birth weight (normal birth weight/high birth weight/low birth weight) and the linear birth weight actually provide the same information. The categorical birth weight cannot thus be regarded as a confounding factor. However, we have added a sensitivity analysis, which may help to answer your questions about how birth weight contributed to the prevalence of MetS. However, the low birth weight group was not comparable because of the small sample size (n=89). The sensitivity analysis results have been included in the paper:
Results:
“The results of the sensitivity analysis showed that breastfeeding duration (longer than 6 months) was a protective factor for MetS (Supplementary Table 1).” (page 8, lines 249-251)
3) How many were preterm babies? Do you have that info? I also find this to be relevant, although not as important as the high or low birth weight. Generally preterm babies are born with low birth weight anyway, and this has an effect on the development of metabolic abnormalities later in life.
Birth weight data were available in our study, but no information regarding the week of pregnancy when childbirth occurred. As observed by the reviewer, low birth weight is the variable that may have an impact on the development of metabolic abnormalities later in life.
4) In table 2, you presented numbers of participants who had 0, 1, 2 or 3 or more components of the MetS and then those who had MetS... I am confused because, based on the paper published in circulation in 2009 (Alberti et al.) harmonizing the metabolic syndrome, it has been agreed that ANYONE having 3 or more components IS classified as having MetS. How is it possible that you have different numbers for those having 3 or more components and those with a full diagnosis of MetS??? Is this different in adolescents somehow??
This study used the IDF definition (2007) in children and adolescents. The metabolic syndrome can be diagnosed by central obesity, plus the presence of two or more clinical features (i.e., elevated triglycerides, low HDL-cholesterol, high blood pressure, or increased plasma glucose). Central obesity, as assessed by waist circumference, was regarded as essential, because of the strength of the evidence linking waist circumference to cardiovascular disease and the other metabolic syndrome components. As a result, a diagnosis of MetS was not equal to three or more components.
5) Finally, you discuss how there are lower incidences (OR) of hypertriglyceridemia in those being breastfed but, in one of the erlier tables, you clearly show a positive correlation in Chinese children between TGs and being breastfed (statistically significant). How do you interpret the contradictory results?
Table 3 only provides preliminary results, which show a possible correlation between TGs and materinal breastfeeding without taking other confounders into account. Howerver, the conclusion should be based on the final regression results because this analytical model is more comprehensive.
Minor comments:
In the method section (2.2), lines 111 and 115 (now line 126), you describe the Chinese and Spanish methodology for measuring body weight; in one you say to the nearest 0.1kg and the other you say to the nearest 100g. This is the same, so I suggest keeping consistency.
The text has been modified as suggested by the reviewer (now line 126)
[1] Zimmet P, Alberti G, Kaufman F, et al. The metabolic syndrome in children and adolescents. Lancet 2007;369:2059-2061.
[2] Brockway M, Benzies K, Hayden KA. Interventions to Improve Breastfeeding Self-Efficacy and Resultant Breastfeeding Rates: A Systematic Review and Meta-Analysis. Journal of human lactation : official journal of International Lactation Consultant Association 2017;33:486-499.
Round 2
Reviewer 1 Report
Thank you for you revision, and it makes it comprehensible.
Lines 44-45
Which population is used in this reference [5], Spanish, or not? If the latter, it is unlikely necessary.
Lines 44-51
Compare the definitions of MetS in theses studies (ref. 6, 7, and 8).
Lines 55-56
When this reference is a cross-sectional study, as the authors responded, this sentence is a speculation without speculation.
Lines 82-83 and 92-93
Repetition. Delete the former. In addition, although the authors mentioned exclusion criteria exclusively in the Chinese population (Lines 92-93), this exclusion criteria might be applied to the Spanish population.
Line 121
BMI [weight (kg)/ {height (m)}2], superscript position.
Line 126
0.1g -> 0.1 kg
Line 168
I requested the authors to mention the cutoffs (unit, cm, and percentiles) of each ethnicity-age-gender group in the before comment. It is useful for readers and researchers who do met-analysis.
Table 1.
P values are better, as<0.01.< p="">
Lines 22-207
The authors responded to the before comment, and compared SDs. I requested the authors to analyze additionally the subjects without extremely high TG in the before comment (sensitivity analysis). And state a limitation.
Lines 212-215
Not to confuse meanings of “strong, or weak” between strength of coefficients and comparison, do not use “strong, or weak” here. The correlation coefficient values indicate weak associations according to Cohen.
Table 3.
When the variable MetS is binary, correlation coefficients are inappropriate and confused for readers. A t-test is appropriate.
Lines 246-248
I requested the authors to clearly describe the directions.
Lines 249-251
The purpose and analysis method of a sensitivity analysis should be explained. Supplementary files are inappropriately uploaded, so I cannot see it.
Table 5.
Explain the effects of interaction terms. Positive, or negative, and effects of each term (breastfeeding duration)
Discuss the effects of breastfeeding duration different between the Chinses and Spanish subjects (opposite direction). This difference may be derived from the range of breastfeeding duration, adiposity extent, or genetic/ethnic traits. Furthermore, interaction terms are significant. If so, this conclusion (Lines 348-355) must not be applied to both populations.
Thank you for you revision, and it makes it comprehensible.
Lines 44-45
Which population is used in this reference [5], Spanish, or not? If the latter, it is unlikely necessary.
Lines 44-51
Compare the definitions of MetS in theses studies (ref. 6, 7, and 8).
Lines 55-56
When this reference is a cross-sectional study, as the authors responded, this sentence is a speculation without speculation.
Lines 82-83 and 92-93
Repetition. Delete the former. In addition, although the authors mentioned exclusion criteria exclusively in the Chinese population (Lines 92-93), this exclusion criteria might be applied to the Spanish population.
Line 121
BMI [weight (kg)/ {height (m)}2], superscript position.
Line 126
0.1g -> 0.1 kg
Line 168
I requested the authors to mention the cutoffs (unit, cm, and percentiles) of each ethnicity-age-gender group in the before comment. It is useful for readers and researchers who do met-analysis.
Table 1.
P values are better, as<0.01.< p="">
Lines 22-207
The authors responded to the before comment, and compared SDs. I requested the authors to analyze additionally the subjects without extremely high TG in the before comment (sensitivity analysis). And state a limitation.
Lines 212-215
Not to confuse meanings of “strong, or weak” between strength of coefficients and comparison, do not use “strong, or weak” here. The correlation coefficient values indicate weak associations according to Cohen.
Table 3.
When the variable MetS is binary, correlation coefficients are inappropriate and confused for readers. A t-test is appropriate.
Lines 246-248
I requested the authors to clearly describe the directions.
Lines 249-251
The purpose and analysis method of a sensitivity analysis should be explained. Supplementary files are inappropriately uploaded, so I cannot see it.
Table 5.
Explain the effects of interaction terms. Positive, or negative, and effects of each term (breastfeeding duration)
Discuss the effects of breastfeeding duration different between the Chinses and Spanish subjects (opposite direction). This difference may be derived from the range of breastfeeding duration, adiposity extent, or genetic/ethnic traits. Furthermore, interaction terms are significant. If so, this conclusion (Lines 348-355) must not be applied to both populations.
Author Response
Response to Reviewer
This manuscript is a resubmission of "nutrients-532204”. We are thankful to the reviewers for the thorough way they looked at our manuscript, which will certainly help to improve its quality. The manuscript has been revised in consonance with the comments and recommendations and we have answered to queries raised by the reviewer 1.
A detailed reply including the reviewer’s comments and our responses and the actions taken is attached, with the reviewer’s comments followed by our response (in italics) and changes.
Comments and Suggestions for Authors
-Lines 44-45. Which population is used in this reference [5], Spanish, or not? If the latter, it is unlikely necessary.
In the study by Miller et al. (Metab. Syndr. Relat. Disord. 2014, 12, 527-532), participants were recruited in the USA only, therefore, and following the reviewer’s recommendation, we have removed the sentences and references from the text.
- Lines 44-51. Compare the definitions of MetS in theses studies (ref. 6, 7, and 8).
In the three studies MetS was defined according to the criteria proposed by the International Diabetes Federation for children and adolescents. A sentence on this has been included in the text. Page 2, lines 57-59.
- Lines 55-56. When this reference is a cross-sectional study, as the authors responded, this sentence is a speculation without speculation.
The sentence has been modified to a less speculative one: “According to these studies carried out in China and Spain it might be expected that MetS would be less frequent in Chinese adolescents”. Page 2, lines 60-61.
- Lines 82-83 and 92-93. Repetition. Delete the former. In addition, although the authors mentioned exclusion criteria exclusively in the Chinese population (Lines 92-93), this exclusion criteria might be applied to the Spanish population.
The former sentence has been deleted and the exclusion criteria have been reformulated to include both Chinese and Spanish adolescents. Page 3, lines 108-112.
- Line 121. BMI [weight (kg)/ {height (m)}2], superscript position.
The superscript position has been corrected. Page 5, line 143.
- Line 126. 0.1g -> 0.1 kg.
The error in the units has been corrected. Page 5, lines 149.
- Line 168. I requested the authors to mention the cutoffs (unit, cm, and percentiles) of each ethnicity-age-gender group in the before comment. It is useful for readers and researchers who do met-analysis.
Thanks for your suggestion. We have added the more detailed information for the cutoffs in the manuscript as:
“(defined by WC adult ethnicity-specific cutoffs: ≥2350px in men and ≥2000px in women for the Spanish group; ≥90 cm in men and ≥80 cm in women for the Chinese group)”. Page 6, lines 199-201.
- Table 1. P values are better, as<0.01.< p="">
0.00 values have been substituted with<0.01.< em="">
- Lines 22-207. The authors responded to the before comment, and compared SDs. I requested the authors to analyze additionally the subjects without extremely high TG in the before comment (sensitivity analysis). And state a limitation.
Thanks for your careful checking. As your comments, we checked the outliers thoroughly based on the Pauta criterion including WC, HDL-C and TG, but only 2 extra outliers were excluded. Then we re-analysis the mean and SD of the relevant variables, and similar results were found. We have updated some results in tables 1 and 2 and we have added a limitation to the corresponding paragraph. Page 14, lines 418-419.
- Lines 212-215. Not to confuse meanings of “strong, or weak” between strength of coefficients and comparison, do not use “strong, or weak” here. The correlation coefficient values indicate weak associations according to Cohen.
We agree that the text was confusing as it mixed comparisons between groups and strength of correlations. We believe that it is much clearer now as follows: Generally, the correlations of breastfeeding duration or birth weight with MetS and its components (Table 3) differed in Chinese and Spanish individuals. In Chinese adolescents correlations were weak. Page 8, lines 246-249.
- Table 3. When the variable MetS is binary, correlation coefficients are inappropriate and confused for readers. A t-test is appropriate.
Thanks for pointing this out. We have deleted the MetS in Table 3. However, we did not do the t-test because of the unbalanced sample size in two groups (30 subjects in MetS group and more than 2000 subjects in non-MetS group). Meanwhile, the regression results in table 4 have indicated the associations of MetS with Breastfeeding duration and Birth weight after taking other confounders into account.
- Lines 246-248. I requested the authors to clearly describe the directions.
We are sorry for missing this request in the previous revision. We have now indicated the directions of the associations in the total sample as follows: For the early-life factors, breastfeeding duration was negatively associated with hypertriglyceridemia (OR 0.87), low HDL-C (OR 0.81), hyperglycemia (OR 0.60) and MetS (OR 0.74), whereas higher birth weight was positively associated with MetS components such as hyperglycemia (OR 1.96) and abdominal obesity (OR 1.15) in the total sample. Page 10, lines 285-289.
- Lines 249-251. The purpose and analysis method of a sensitivity analysis should be explained. Supplementary files are inappropriately uploaded, so I cannot see it.
According to your previous comments, we did the sensitivity analysis to assess the non-linear logistic regression, in which analysis we divided the continuous breastfeeding duration to two groups (≤6 month and >6 month) and divided the birthweight to three groups(normal weight, high birth weight and low birth weight). The results of the sensitivity analysis showed that breastfeeding duration (> 6 months) was a protective factor for MetS (Supplementary Table 1). Please check out the Supplementary Table 1 as below:
Supplementary Table 1. Risk of MetS features based on associated factors from multivariable logistic regression (categorical variables).
Categorical Variables (reference) | N | MetS |
OR(95%CI) | ||
Age | 1.01 (0.80,1.29) | |
Gender (Boy) | 1011 | 1.48 (0.69,3.19) |
Country (China) | 1150 | 14.52 (5.53,38.13) |
Breastfeeding duration (≤6 month) | 732 | 0.50 (0.20,0.89) |
Birth weight (normal weight) | 1720 | |
High birth weight(>4kg) | 193 | 1.24 (0.46,3.36)
|
Low birth weight(<2.5kg)< span=""> | 89 | NA |
- Table 5. Explain the effects of interaction terms. Positive, or negative, and effects of each term (breastfeeding duration).
We have increased the detail in the explanation of Table 5, by including information on the effects of interaction between variables. The resulting paragraph is as follows: We also performed analyses to assess the interaction of breastfeeding duration and the country of origin and birth weight with the country of origin, using China as reference. When the interaction of breastfeeding duration and the country of origin was assessed, we observed that the effect was significant for low HDL-c, hyperglycemia and hypertriglyceridemia. In contrast, the interaction of birth weight with the country of origin was significant only for hyperglycemia. Page 10, lines 289-295.
We are often confronted in public health by associations that vary by population or subpopulation. In the current study, we used the interaction terms to detect whether country of origin was an effect modifier in the associations of MetS features with the breastfeeding and birthweight. We can not decide the direction of the effects directly from regression results, but the p-value help to shed light on the interaction effect for further study. More detailed information about the interaction term and effect modifier could be accessed in the follow reference: Lopez, P. M., et al. (2019). "Effect measure modification conceptualized using selection diagrams as mediation by mechanisms of varying population-level relevance." J Clin Epidemiol 113: 123-128
- Discuss the effects of breastfeeding duration different between the Chinses and Spanish subjects (opposite direction). This difference may be derived from the range of breastfeeding duration, adiposity extent, or genetic/ethnic traits. Furthermore, interaction terms are significant. If so, this conclusion (Lines 348-355) must not be applied to both populations.
We have included a paragraph in the discussion section about the factors influencing the effect of breastfeeding duration in the differences observed between Chinese and Spanish subjects. Page 13, lines 347-361.
Also, we have modified the conclusions to remark the differences observed between the two countries regarding the effects of breastfeeding duration. Page 15, lines 438-441.
We are thankful to the reviewer for his/her help to improve the manuscript.
Reviewer 2 Report
I believe that the issues raised were addressed adequately, including some extra steps in the analysis of results.
Author Response
We greatly appreciate the comments of the reviewer.